Association between lymphocyte-to-monocyte ratio and stroke-associated pneumonia: a retrospective cohort study

Li Xiaoqiang liboleq@163.com 1
Zhou Xiangmao 2
Wang Hui 1
Ruan Baifu 1
Song Zhibin 1
Zhang Guifeng 1
1 Department of Neurology, Xiaolan People’s Hospital of Zhongshan (The Fifth People’s Hospital of Zhongshan) , Zhongshan , Guangdong , China
2 Department of Gastrointestinal Surgery, The Central Hospital of Yongzhou , Yongzhou , Hunan , China
Barbosa Neto Octavio
Electronic publication date: 2024 Sep 19
Publication date: 2024
Volume: 12
Electronic Location ID: e18066
Received 2024 May 17; Accepted 2024 Aug 19
Copyright: ©2024 Li et al.
Copyright year: 2024
Copyright holder: Li et al.
License: This is an open access article distributed under the terms of the Creative Commons Attribution License, which permits unrestricted use, distribution, reproduction and adaptation in any medium and for any purpose provided that it is properly attributed. For attribution, the original author(s), title, publication source (PeerJ) and either DOI or URL of the article must be cited.
License URL: https://creativecommons.org/licenses/by/4.0/

Keywords: Stroke-associated pneumonia, Lymphocyte-to-monocyte ratio, Acute ischemic stroke, Retrospective cohort study, China

Funding: The Zhongshan Science and Technology Innovation Commission 2022B1097 Our study was supported by The Zhongshan Science and Technology Innovation Commission (grant number 2022B1097). The funders had no role in study design, data collection and analysis, decision to publish, or preparation of the manuscript.

==============================
Background

Stroke-associated pneumonia (SAP) is a common complication of acute ischemic stroke (AIS) and is associated with increased mortality and prolonged hospital stays. The lymphocyte-to-monocyte ratio (LMR) is a novel inflammatory marker that has been shown to be associated with various diseases. However, the relationship between the LMR and SAP in patients with AIS remains unclear.

Methods

A retrospective cohort study was conducted on 1,063 patients with AIS admitted to our hospital within 72 hours of symptom onset. Patients were divided into two groups: the SAP group (n = 99) and the non-SAP group (n = 964). The LMR was measured within 24 hours of admission, and the primary outcome was the incidence of SAP. We used univariate and multivariate logistic regression analyses to assess the relationship between the LMR and SAP. Additionally, curve-fitting techniques and subgroup analyses were conducted.

Result

The incidence of SAP was 9.31%. We found that the LMR was significantly lower in the SAP group than in the non-SAP group (2.46 ± 1.44 vs. 3.86 ± 1.48, P < 0.001). A nonlinear relationship was observed between the LMR and the incidence of SAP. Subgroup analysis revealed that an elevated LMR was associated with a reduced incidence of SAP in individuals with an LMR below 4. Multivariate logistic regression analysis demonstrated that LMR was an independent predictor of SAP (OR = 0.37, 95% CI [0.27–0.53]).

Conclusion

Our study suggests that the LMR is an independent predictor of SAP in patients with AIS, particularly when the LMR is less than 4. The LMR may serve as a promising biomarker for the early identification of patients with AIS at a high risk of SAP.

Introduction

Stroke is the leading cause of death and disability worldwide, with ischemic stroke being the most prevalent type (Feigin et al., 2022). Stroke-associated pneumonia (SAP) is a common complication of acute ischemic stroke (AIS), with a reported incidence varying between 3.2% and 56.6% in different settings (Hannawi et al., 2013; Bruening & Al-Khaled, 2015; Liu et al., 2022). SAP frequently results in unfavorable functional outcomes following AIS, contributing to increased mortality, morbidity, and prolonged hospital stays (Hannawi et al., 2013; Tinker et al., 2021). Therefore, the early detection and effective prevention of SAP are of paramount importance. Consequently, it is essential to investigate the risk factors and measurable biomarkers of SAP in patients with AIS.

A growing body of evidence has highlighted the critical role of the neuroinflammatory response in the pathophysiology of ischemic stroke (Yang et al., 2019; Candelario-Jalil, Dijkhuizen & Magnus, 2022; Westendorp et al., 2022). The inflammatory response is a complex process involving various cells and molecules, including leukocytes, cytokines, chemokines, and adhesion molecules. A neuroinflammatory reaction may lead to blood–brain barrier disruption, neuronal damage, and cerebral edema, ultimately resulting in poor outcomes after stroke. Recently, the lymphocyte-to-monocyte ratio (LMR) has gained attention as a novel inflammatory marker that potentially reflects baseline inflammation. The LMR is calculated by dividing the lymphocyte count by the monocyte count. Previous studies have demonstrated that the LMR is associated with a range of diseases, including coronary artery disease, cancer, and atherosclerotic stenosis (Kose et al., 2019; Mandaliya et al., 2019; Wu et al., 2022). In patients with AIS, LMR is linked to prognosis, functional outcomes, and hemorrhagic transformation after intravenous thrombolysis (Song et al., 2021; Gong et al., 2021; Li et al., 2022).

Research on the relationship between the LMR and SAP is still in its preliminary stages. Some studies have indicated a negative correlation between serum LMR and SAP, suggesting that LMR may be a useful tool for identifying patients with AIS at high risk of SAP (Ren et al., 2017a; Cheng et al., 2020; Cao et al., 2021). However, these studies have certain limitations, including small sample sizes and a lack of exploration of the nonlinear relationship between LMR and SAP. To address these limitations, we conducted this study to investigate the relationship between the LMR and SAP in patients with AIS.

Methods

Study design and participants

We conducted a retrospective cohort study of patients diagnosed with AIS and admitted to our hospital who were consecutively recruited to minimize selection bias. The following inclusion criteria were applied: (1) age ≥ 18 years; (2) diagnosis of AIS according to the World Health Organization criteria; (3) complete laboratory data within 24 h of admission; and (4) admission within a 72-hour window post-stroke onset. Patients were excluded if they had any of the following: (1) history of severe liver or kidney dysfunction; (2) malignant tumors; (3) immunodeficiency diseases; or (4) use of immunosuppressive agents.

Data collection

We collected demographic data, medical histories, laboratory findings, and clinical characteristics from the electronic medical records of all participants. The collected variables included age, sex, and history of hypertension, diabetes mellitus, and atrial fibrillation. Blood samples for complete blood count parameters, including white blood cells, neutrophils, lymphocytes, and monocytes, were obtained immediately upon admission. Additional blood samples for lipid profiles (TG, HDL-C, LDL-C, and TC) were collected within 24 h of admission. We calculated the LMR by dividing the lymphocyte count by the monocyte count. After removing outliers in the LMR data (18 of 1,081), our final sample comprised 1,063 consecutive AIS patients admitted from October 2019 to November 2022. The patient selection process is illustrated in Fig. 1. This study was approved by the Ethics Committee of Xiaolan People’s Hospital of Zhongshan (approval number: 2022-0026). The requirement for informed patient consent was waived due to the retrospective nature of the study.

Figure 1 Flow chart visualizing the patient selection process.

Definition of stroke-associated pneumonia

We defined SAP according to the standardized criteria outlined in the 2015 consensus (Smith et al., 2015). This definition involves a combination of clinical evaluations and laboratory examinations that were further corroborated by retrospective analyses of sputum cultures and chest CT scans from the patients’ medical records.

Statistical analysis

Continuous variables were expressed as mean ± standard deviation or median (interquartile range) and compared using the Student’s t-test or Mann–Whitney U test, as appropriate. Categorical variables were expressed as percentages and compared using the chi-square or Fisher’s exact test. Curve fitting analysis was performed to investigate the relationship between LMR and the incidence of SAP. This method allows for the exploration of both linear and non-linear relationships. We employed univariate and multivariate Cox proportional hazard models to evaluate the relationship between the LMR level and the risk of SAP. Three distinct models were implemented: The first model (Model 1, unadjusted) was created without any adjustments. The second model (Model 2) was created with adjustments for age and sex. The third model (Model 3, adjusted for a multitude of factors including sex, age, length of hospital stay, and medical conditions such as diabetes, hypertension, atrial fibrillation, triglycerides, high-density lipoprotein cholesterol, total cholesterol, low-density lipoprotein cholesterol, neutrophils, lymphocytes, monocytes, and white blood cells) was created with adjustments for a multitude of factors. Statistical significance was set at p < 0.05. All statistical analyses were conducted using the R statistical software (https://www.R-project.org, The R Foundation) and EmpowerStats (http://www.empowerstats.com, X&Y Solutions, Inc., Boston, MA, USA).

Results

Baseline characteristics

A total of 1,063 patients with AIS were included in this study. The incidence of SAP was 9.3% (99/1063). The baseline characteristics of patients with and without SAP are summarized in Table 1. Patients with SAP were older and had a higher prevalence of atrial fibrillation and venous thrombolysis (P < 0.05). Moreover, white blood cell count, neutrophil count, monocyte count, and HDL-C levels were significantly elevated in the SAP group compared to the non-SAP group. Conversely, lymphocyte counts were significantly lower in the SAP group (P < 0.05). Consequently, the LMR was significantly lower in the SAP group compared to the non-SAP group (2.46 ± 1.44 vs 3.86 ± 1.48, P > 0.001). No significant differences were observed in the remaining parameters (P > 0.05).

Table 1 Baseline characteristics of participants according to stroke-associated pneumonia.

	Non-SAP (964)	SAP (99)	P-value	
Sex (male)	659 (68.36%)	66 (67.68%)	0.730c	
Age, years	61.65 ± 12.68	68.37 ± 14.24	<0.001a	
Hospitalization days	9.69 ± 5.15	13.54 ± 6.79	<0.001a	
WBC, 10 ˆ9/L	8.31 ± 2.82	9.90 ± 3.95	<0.001b	
Lymphocytes, 10 ˆ9/L	1.62 ± 0.61	1.24 ± 0.58	<0.001b	
Neutrophils, 10 ˆ9/L	6.04 ± 2.77	7.93 ± 3.76	<0.001b	
Monocyte	0.46 ± 0.18	0.59 ± 0.34	<0.001b	
Lymphocyte-to-monocyte ratio (LMR)	3.86 ± 1.48	2.46 ± 1.44	<0.001b	
TG, mmol/L	1.74 ± 1.27	1.49 ± 1.12	0.069b	
HDL-C, mmol/L	1.08 ± 0.28	1.15 ± 0.38	0.042a	
LDL-C, mmol/L	3.08 ± 0.95	3.03 ± 1.21	0.615a	
TC, mmol/L	4.64 ± 1.09	4.58 ± 1.32	0.611a	
Diabetes	350 (36.31%)	32 (32.32%)	0.431c	
Hypertension	812 (84.23%)	86 (86.87%)	0.490c	
Atrial fibrillation	48 (4.98%)	18 (18.18%)	<0.001c	
Venous thrombolysis	120 (12.45%)	29 (29.29%)	<0.001c	
Notes.

Continuous data are shown as mean ± SD (normal distribution) or median (quartile) (skewed distribution). Categorical data are shown as n (%). Statistical methods: Continuous variables were compared using Student’s t-test for normally distributed data or Mann-Whitney U test for skewed data. Categorical variables were compared using Chi-square test or Fisher’s exact test as appropriate.

a Student’s t-test.

b Mann–Whitney U test.

c Chi-square test or Fisher’s exact test.

WBC White blood cells

TG Triglyceride

TC Total cholesterol

LDL-C Low-density lipoprotein cholesterol

HDL-C High-density lipoprotein cholesterol

SAP stroke-associated pneumonia

Figure 2 The relationship between lymphocyte-to-monocyte ratio level and the incidence of stroke-associated pneumonia.

Solid line represents the smooth curve fit between variables. Dotted line represents the 95% of confidence interval from the fit. All adjusted for: gender, age, length of hospital stay, and medical conditions such as diabetes, hypertension, and atrial fibrillation.

Relationship between LMR and SAP incidence in AIS patients

We aimed to characterize the relationship between LMR and the incidence of SAP using curve fitting. After adjusting for sex, age, length of hospital stay, and medical conditions (diabetes, hypertension, and atrial fibrillation), we found that the relationship between LMR and SAP was nonlinear (Fig. 2). We applied a two-piecewise linear regression model to identify the inflection point of the LMR, which was 4 (log-rank test, p > 0.05; Table 2). Below the inflection point, a negative correlation was observed between the LMR and the incidence of SAP (OR = 0.41, 95% CI [0.31–0.53], p > 0.01). Conversely, a saturation effect was observed above the inflection point (OR = 1.03, 95% CI [0.67–1.57], p = 0.9062; Table 2). Based on these findings, to investigate the negative association between LMR and the incidence of SAP, we selected patients with an LMR below 4.

Univariate analysis of covariates and SAP incidence (LMR <4)

We performed a univariate analysis to identify factors associated with SAP occurrence (Table 3). Without adjusting for potential confounders, the analysis revealed several variables significantly associated with SAP incidence. LMR (OR = 0.30, 95% CI [0.22–0.40]), age (OR = 1.04, 95% CI [1.02–1.06]), length of hospital stay (OR = 1.10, 95% CI [1.06–1.14]), white blood cell (WBC) count (OR = 1.10, 95% CI [1.03–1.17]), lymphocyte count (OR = 0.33, 95% CI [0.20–0.56]), neutrophil count (OR = 1.12, 95% CI [1.05–1.19]), monocyte (OR = 7.39, 95% CI [2.90–18.82]), presence of atrial fibrillation (OR = 3.34, 95% CI [1.70–6.57]), and venous thrombolysis (OR = 2.63, 95% CI [1.54–4.48]) were positively associated with the incidence of SAP. Nevertheless, there was no significant correlation observed between SAP and variables such as gender, HDL-C, LDL-C, TG, diabetes, and hypertension.

Multivariate analysis of LMR and SAP incidence (LMR <4)

To elucidate the relationship between LMR and SAP incidence, we performed a multivariate analysis using LMR as the independent variable and the risk of SAP as the dependent variable. This analysis was adjusted for a wide array of variables, including sex, age, length of hospital stay, TG, HDL-C, TC, LDL-C, neutrophils, lymphocytes, monocytes, WBC, and medical conditions, such as diabetes, hypertension, and atrial fibrillation. In Model 1, LMR levels showed a negative correlation with SAP risk (OR = 0.30, 95% CI [0.22–0.40]; Table 4). This positive correlation persisted in the minimally adjusted Model 2 (OR = 0.32; 95% CI [0.24–0.44]). In the fully adjusted Model 3, the negative association remained significant (OR = 0.37, 95% CI [0.27–0.53]). For a more robust analysis, we stratified LMR into tertiles. Compared to the lowest tertile (T1, reference group), both T2 and T3 showed lower odds of SAP incidence across all models, with the highest tertile (T3) consistently demonstrating the lowest risk (Model 3: OR = 0.18, 95% CI [0.07–0.42]).

Table 2 Results of two-piecewise linear regression model.

We adjusted sex and age; length of hospital stay, and medical conditions such as diabetes, hypertension, and atrial fibrillation, TG, HDL-C, TC, LDL-C, neutrophils, lymphocytes, monocyte and WBC.

	Incidence of SAP (OR, 95% CI)	p-value	
Fitting model by standard linear regression	0.52 (0.43, 0.63)	<0.0001	
Fitting model by two-stage linear regression			
The inflection point of LMR	4	
<4	0.41 (0.31, 0.53)	<0.0001	
≥4	1.03 (0.67, 1.57)	0.9062	
p for log likelihood ratio test	0.004	
Notes.

LMR, The lymphocyte-to-monocyte ratio. SAP, Stroke-associated pneumonia.

Table 3 Crude association to identify risk factors associated with stroke-associated pneumonia in AIS patients (LMR < 4).

	Statistics	OR(95% CI)	P- value	
Sex (Male vs. Female)	(462 vs 171)	0.85 (0.50, 1.46)	0.5666	
Age (year)	63.37 ± 13.20	1.04 (1.02, 1.06)	<0.0001	
Hospitalization days	10.34 ± 5.57	1.10 (1.06, 1.14)	<0.0001	
White blood cell	9.02 ± 3.27	1.10 (1.03, 1.17)	<0.0001	
Lymphocytes	1.36 ± 0.52	0.33(0.20, 0.56)	<0.0001	
Neutrophils	6.95 ± 3.21	1.12 (1.05, 1.19)	0.00009	
Monocyte	0.53 ± 0.22	7.39 (2.90, 18.82)	<0.0001	
TG	1.68 ± 1.41	0.82 (0.63, 1.06)	0.1267	
HDL-C	1.10 ± 0.30	1.46 (0.65, 3.25)	0.3597	
LDL-C	2.99 ± 0.98	1.05 (0.82, 1.35)	0.6717	
Lymphocyte-to-Monocyte Ratio	2.71 ± 0.85	0.30 (0.22, 0.40)	<0.0001	
TC	4.54 ± 1.15	1.02 (0.83, 1.26)	0.8553	
Diabetes (without vs. with)	(424 vs. 209)	0.76 (0.46, 1.28)	0.3062	
Hypertension (without vs. with)	(105 vs. 528)	1.50 (0.75, 3.01)	0.2544	
Atrial fibrillation (without vs. with)	(587 vs. 46)	3.34 (1.70, 6.57)	0.0005	
Venous thrombolysis (without vs. with)	(534 vs. 99)	2.63 (1.54, 4.48)	0.0004	
Notes.

OR odds ratio

CI confidence interval

WBC White blood cells

TG Triglyceride

TC Total cholesterol

LDL-C Low-density lipoprotein cholesterol

HDL-C High-density lipoprotein cholesterol

SAP stroke-associated pneumonia

Table 4 Relationship between lymphocyte-to-monocyte ratio level in different models of multivariate analysis (LMR < 4).

Variable	Model 1(OR, 95% CI, P)	model 2(OR, 95% CI, P)	model 3(OR, 95% CI, P)	
Lymphocyte-to-monocyte ratio	0.30 (0.22, 0.40) <0.0001	0.32 (0.24, 0.44) <0.0001	0.37 (0.27, 0.53) <0.0001	
Lymphocyte-to-monocyte ratio (tertiles)				
T1	Ref	Ref	Ref	
T2	0.39 (0.23, 0.66) 0.0005	0.42 (0.24, 0.71) 0.0015	0.53 (0.29, 1.00) 0.0491	
T3	0.11 (0.05, 0.24) <0.0001	0.12 (0.05, 0.28) <0.0001	0.18 (0.07, 0.42) 0.0001	
Notes.

Model 1: no variables are adjusted. Model 2 adjust for: sex and age. Model 3 adjust for: gender, age, length of hospital stay, and medical conditions such as diabetes, hypertension, and atrial fibrillation, TG, HDL-C, TC, LDL-C, neutrophils, lymphocytes, monocyte and WBC.

T1, T2, and T3 represent the first, second, and third tertiles of the lymphocyte-to-monocyte ratio, respectively.

Discussion

The objective of this study was to investigate the association between the LMR and SAP in patients with AIS. Our findings support the potential use of LMR as an accessible and independent predictor of SAP in this population.

To our knowledge, This is the first comprehensive exploration of the LMR-SAP relationship in a large cohort of AIS patients (n = 1063). We observed a significant negative correlation between lower admission LMR and increased SAP risk. These results align with previous studies identifying elevated LMR as a protective factor against infections (Ren et al., 2017a; Guan, Wang & Zhao, 2023). Additionally, the SAP prevalence in our study (9.3%) was consistent with previous reports (Hannawi et al., 2013; Quyet et al., 2019; Westendorp et al., 2022), lending credibility to our findings. Furthermore, our analysis identified several well-established risk factors for SAP, including age, length of hospitalization, and white blood cell and neutrophil counts, thereby corroborating the existing literature (Qiu et al., 2022; Wang et al., 2023; Arsava et al., 2023). The association between LMR and SAP remained statistically significant after adjusting for these confounders, underscoring the independent predictive value of the LMR. Notably, patients with SAP in our study experienced significantly prolonged hospital stays, aligning with Arboix et al. (2012). who identified infections as key predictors of extended hospitalization in acute stroke patients. While their study examined various medical complications, our research specifically highlights LMR as a potential early predictor of SAP risk and, consequently, prolonged hospitalization.

The precise mechanisms underlying SAP development remain unclear. However, a prominent theory suggests that stroke-induced immunosuppression syndrome (SIDS) may be a contributing factor (Liu et al., 2018; Guo et al., 2022). Ischemic stroke triggers an inflammatory cascade, leading to the infiltration of immune cells into the injured brain tissue (Kim, Kawabori & Yenari, 2014; Dylla et al., 2021). This intricate interplay between the immune system and the central nervous system can ultimately culminate in SIDS, rendering patients more susceptible to opportunistic infections such as SAP (Shi et al., 2018; Faura et al., 2021). LMR offers a composite measure of the inflammatory state by combining lymphocyte count, which contributes to immunoregulation, with monocyte count, which influences inflammation (Qi et al., 2018, p. 90). A lower LMR indicates a preponderance of monocytes and a potentially compromised immune response, which may increase the risk of infections, such as SAP. This concept aligns with the protective effect of elevated LMR observed in our study.

The growing recognition of neuroinflammation in AIS pathogenesis highlights the potential utility of the LMR as a prognostic marker (Chamorro et al., 2016; Becker & Buckwalter, 2016; Shi et al., 2019). The LMR can be readily calculated from routine blood tests, making it a cost-effective and accessible tool for clinical use. Our findings add to the growing body of research highlighting the association between the LMR and the prognosis of AIS patients (Ren et al., 2017b; Park et al., 2018; Gong et al., 2021). Notably, a recent study by Oh et al. (2020) showed that the LMR could predict poor functional outcomes and symptomatic intracerebral hemorrhage in patients with AIS treated with mechanical thrombectomy.

Our study identifies the LMR as a promising predictor of SAP in patients with AIS. Its ease of calculation and independent predictive value suggest its potential as a screening tool for early SAP detection. Early identification of high-risk patients could enable implementation of preventive measures, such as meticulous aspiration precautions and respiratory physiotherapy, potentially reducing the incidence of SAP and improving patient outcomes. Moreover, given the potential association between the LMR and post-stroke functional outcomes, it may also serve as a prognostic marker of overall patient recovery. Further research should explore the utility of LMR in various aspects of AIS management, including its potential to guide treatment decisions such as immunomodulatory therapies.

This study has several limitations. The single-center design and relatively modest sample size necessitate validation in more extensive multicenter studies to minimize selection bias and enhance generalizability. Additionally, we only evaluated LMR at admission, neglecting its potential dynamic changes during hospitalization. Future studies exploring serial LMR measurements could provide valuable insights into SAP’s disease progression and risk stratification. Furthermore, our study focused on AIS patients as a whole, without stratification by age groups (Arboix et al., 2000) or ischemic stroke subtypes. Future research should investigate the relationship between LMR and SAP in different age groups and across various ischemic stroke subtypes to provide more nuanced insights. Another limitation is that we did not include in-hospital mortality data or causes of death in our study design. Future research should consider incorporating these outcomes to provide a more comprehensive understanding of the relationship between LMR, SAP, and patient outcomes. Finally, we did not evaluate other inflammatory markers or potential confounders, such as cytokine levels. Future studies incorporating a more comprehensive assessment of the inflammatory milieu could offer a deeper understanding of the complex interplay between inflammation and SAP development.

Conclusion

In conclusion, our study reveals that the LMR is an independent predictor of SAP in patients with AIS. This readily available and easily calculated marker could be used as a screening tool for early SAP detection and may have prognostic implications for overall patient recovery. Further research is warranted to explore the broader utility of LMR in AIS management and to elucidate the underlying mechanisms linking LMR to SAP development.

Supplemental Information

Supplemental Information 1 STROBE checklist

Supplemental Information 2 Codebook

Data S1 Raw data

Abbreviations

AIS acute ischemic stroke

HDL-C high-density lipoprotein cholesterol

LDL-C low-density lipoprotein cholesterol

LMR lymphocyte-to-monocyte ratio

SAP stroke-associated pneumonia

TC total cholesterol

TG triglyceride

WBC white blood cell count

Additional Information and Declarations

Competing Interests

Author Contributions

Human Ethics

Field Study Permissions

Data Availability

The authors declare there are no competing interests.

Xiaoqiang Li conceived and designed the experiments, analyzed the data, prepared figures and/or tables, authored or reviewed drafts of the article, and approved the final draft.

Xiangmao Zhou performed the experiments, analyzed the data, authored or reviewed drafts of the article, and approved the final draft.

Hui Wang performed the experiments, authored or reviewed drafts of the article, and approved the final draft.

Baifu Ruan analyzed the data, prepared figures and/or tables, and approved the final draft.

Zhibin Song conceived and designed the experiments, analyzed the data, prepared figures and/or tables, authored or reviewed drafts of the article, and approved the final draft.

Guifeng Zhang analyzed the data, authored or reviewed drafts of the article, and approved the final draft.

The following information was supplied relating to ethical approvals (i.e., approving body and any reference numbers):

This study was approved by the Ethics Committee of People’s Hospital.

The following information was supplied relating to field study approvals (i.e., approving body and any reference numbers):

This study was approved by the Ethics Committee of People’s Hospital (project number:2022-0026).

The following information was supplied regarding data availability:

The raw measurements are available in the Supplementary File.

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
