# Peer review of "Association between lymphocyte-to-monocyte ratio and stroke-associated pneumonia: a retrospective cohort study"

_PeerJ, doi:10.7717/peerj.18066_

## Round 0.1 · original submission · Major Revisions

Dear authors,

The study entitled “Association between lymphocyte-to-monocyte ratio and stroke-associated pneumonia: A retrospective cohort study” demonstrated interesting findings using an appropriate methodological approach. However, some important points must be clarified in the manuscript. Your article has great potential for publication on PeerJ, but the reviewers have requested substantial changes to be made.

Please ensure that all review, editorial, and staff comments are addressed in a response letter and that any edits or clarifications mentioned in the letter are also inserted into the revised manuscript where appropriate.

Reviewer 1 ·

Basic reporting

1. The language needs to be reworked.
2. Statistical methods need to be marked in Table 1.

Experimental design

1. in the section of "Study Design and Participants", patients diagnosed with AIS and admitted to our hospital from January 2018 to December 2022 were consecutively recruited to minimize selection bias. But, in the section of "Data Collection" ,Based on these criteria, we retrospectively included 1063 consecutive AIS patients in this study, from October 2019 to November 2022. Why was the time difference so much?
2. When was the laboratory data collected?
3. Who would determined whether the patient was stroke-associated pneumonia? and when?

Validity of the findings

There were problems with the design and it was impossible to judge the validity of the results

Additional comments

no comment

·

Basic reporting

The authors present the results of a retrospective single-center cohort clinical study aimed to analyze the relationship between the lymphocyte-to-monocyte ratio (LMR) and stroke-associated pneumonia (SAP) in patients with acute ischemic stroke (AIS). A total of 1063 patients with AIS were enrolled in this study. The authors found that the incidence of SAP was 9.31%. showing that LMR may serve as a promising biomarker for the early identification of patients with AIS at a high risk of SAP. The study is potentially interesting, but can improved if the fo-llowing considerations are addressed:

Experimental design

No comment

Validity of the findings

No comment

Additional comments

1.It would be interesting to include in the text a comment regarding that in a clinical study, in-hospital medical complications (including pneumonia) were relevant factors influencing the dura-tion of hospitalization after acute stroke (Int J Clin Med 2012; 3 : 502-507). See and comment on this reference. Was this clinical situation confirmed in the study sample?

2.What was the in-hospital mortality rate in the study sample?

3.It would be useful to mention the frequency of in-hospital mortality between SAP and non-SAP patients and the causes of death (neurological and non-neurological) in the study sample.

4.It would be interesting to know the different ischemic stroke subtypes in the study population.

5.It would be interesting for the authors to discuss, as a new line of research to analyze the results obtained in the subgroup of very old patients (85 years old or more) given that demographics and risk factors are quite different in this age segment of stroke patients (see and comment the study published in Acta Neurol Scand 2000; 101: 25-29).

6.A brief concluding comment on other possible lines of future research on the presented topic would be appreciated

---

## Round 0.2 · Minor Revisions

Dear authors,

Although the authors made significant changes to the manuscript “Association between lymphocyte-to-monocyte ratio and stroke-associated pneumonia: A retrospective cohort study”, one of the reviewers still asks how you calculated the sample size.

Reviewer 1 ·

Basic reporting

no

Experimental design

How is the sample size calculated?

Validity of the findings

no

·

Basic reporting

The authors have made the previously recommended changes and responded the criticisms. Thank you for making these changes.

Experimental design

Rigorous investigation

Validity of the findings

Data are robust.

Additional comments

No comments

---

## Round 0.3 · accepted · Accept

Dear Author,

Congratulations! After your diligent work addressing the reviewers' comments, I am pleased to inform you that your manuscript has been accepted for publication in PeerJ. This version is more concise and formal, enhancing clarity and flow.